# Effects of Microbial Fertilizer on Soil Fertility and Alfalfa Rhizosphere Microbiota in Alpine Grassland

Yangan Zhao [1], Guangxin Lu [1,*], Xin Jin [1], Yingcheng Wang [1], Kun Ma [1], Haijuan Zhang [1], Huilin Yan [1] and Xueli Zhou [2]

[1] Collage of Agriculture and Animal Husbandry, Qinghai University, Xining 810016, China; 18893093541@163.com (Y.Z.); 18894310895@163.com (X.J.); yingcheng_w@163.com (Y.W.); m18294258973@163.com (K.M.); 15297196583@163.com (H.Z.); yan06112021@163.com (H.Y.)
[2] Grassland Improvement Experiment Station, Gonghe 813000, China; zhouxuelia@163.com
\* Correspondence: lugx74@163.com; Tel.: +86-13897216290

**Abstract:** Chemical fertilizers are gradually being replaced with new biological fertilizers, which can improve the soil and soil microorganisms. In this experiment, leguminous forage (*Medicago sativa* cv. Beilin 201) was used as the research object. By measuring alfalfa root systems and soil properties and using high-throughput sequencing technology, we investigated the effect of biological (rhizobial) fertilizer at different concentrations on soil fertility and alfalfa rhizosphere microbiota in alpine grasslands. The results demonstrated that the treatment with biofertilizer significantly reduced total nitrogen (TN) and total organic carbon (TOC) content in soils, increased root densities, and significantly increased the number of root nodules in alfalfa. There were differences in the response of rhizosphere microorganisms to different concentrations of biofertilizer, and the treatment with biofertilizer led to pronounced changes in the microbial community structure. The abundance of beneficial bacteria such as *Rhizobium*, *Arthrobacter*, and *Pseudomonas* was significantly increased. The Pearson correlation analysis showed that soil moisture and soil conductivity were significantly positively correlated with the observed richness of rhizosphere microbiota ($p < 0.05$). Meanwhile, Actinobacteria showed a significantly positive correlation with nitrate, TOC, and TN ($p < 0.01$). These results indicated that biofertilizers enhanced soil fertility and altered the rhizosphere microbiota of alfalfa in alpine grassland.

**Keywords:** upland region; biofertilizer; soil fertility; root system rhizomes; inter-root microbial community structure; Pearson correlation analysis

## 1. Introduction

Medicago sativa, a high-quality leguminous plant also known as the "king of pasture", plays an important role in agriculture and animal husbandry [1–4]. Currently, the United States and Russia lead the world in the process of alfalfa industrialization [5]. Alfalfa's high feed value, economic value, and soil improvement value are highly valued and sought after by various countries [6]. In recent years, biofertilizer has gradually replaced chemical fertilizer and become an indispensable part of agriculture [7]. A biofertilizer is a special type of fertilizer that contains a large number of living microorganisms [8]. When biofertilizer is applied to the soil under suitable conditions, these live microorganisms are active and can multiply around the roots of the crop and facilitate autogenous nitrogen fixation or combined nitrogen fixation [9,10]. We found that the application of biofertilizers promoted the decomposition of phosphorus and potassium minerals in the soil, which encouraged crops to absorb or secrete growth hormones to further stimulate crop growth.

Rhizobium, a class of Gram-negative bacteria [11,12], can, under suitable conditions, induce the cortical cells of legume roots and stems to proliferate and form nodules and can form a large number of symbiotic nitrogen-fixing mycobacteria in the root nodules for

protein synthesis in plants [13–15]. However, due to poor environmental and climatic conditions, coupled with the low number of locally bred finished varieties and more serious grass degradation [16] in the Grassland Improvement Experiment Station of Qinghai Province, its quality and adaptability performance varies, which affects its yield and quality [17,18]. There are consistent barriers to alfalfa planting, mainly poor yield, quality decline, soil fertility decline, soil quality degradation, etc. [19,20]. Microorganisms and soil ecological functions are complementary and closely related, and there is a close relationship between plant inter-rooted soil and plants [21,22]. The composition of microbial communities and their diversity is an important indicator of soil properties and functions. Studying the similarities and differences in alfalfa disease formation in alpine grassland can provide a theoretical basis for the scientific planting of alfalfa and promote the rapid development of animal husbandry with high economic and ecological benefits [23,24].

We fully considered the need for legume forage yield and quality in the alpine zone, with the trend of biofertilizers gradually replacing traditional chemical fertilizers. Different gradients of biofertilizers were applied to artificially established alfalfa grasses at the Qinghai Grassland Improvement Experiment Station in China. Plant roots and soil physicochemical properties were determined, and high throughput techniques were employed to determine microorganisms in the inter-rhizosphere soil of the forage, which was investigated for soil fertility and microbial community structure and diversity. We hypothesized that (I) biofertilizers could significantly change the morphological structure of forage roots, (II) biofertilizers can affect soil physical and chemical properties and fertility, and (III) biofertilizers can change the structure and diversity of microbial communities.

## 2. Methods and Materials

### 2.1. Experimental Material

Plant: Alfalfa (Medicago sativa CV. Beilin 201).

Biofertilizer: Nitrogen-fixing rhizobia fertilizer (25 kg/bag) was obtained from Sichuan Grass Ecological Technology Development Co. The strain was rhizobium, mainly including rhizobia and bradyrhizobia. Both genera belong to the order rhizobia.

### 2.2. Overview of the Study Area

The Grassland Improvement Experiment Station (Figure 1) of Qinghai Province, China, is located on the west coast of Qinghai Lake, east of Qinghai Lake, 99°35′ E, 37°05′ N, 3270 m above sea level. The average annual temperature is −0.7 °C, the average temperature of the hottest month (July) is 17.5 °C, the average temperature of the coldest month (January) is −22.6 °C, and the extreme temperature is −34.3 °C. The average frost-free period is 78.7 days, with no absolute frost-free period. With 2670 h of sunshine, 1331.3 °C annual cumulative temperature $\geq 0$ °C, 368.11 mm annual precipitation, 1495.3 mm annual evaporation, and 58% relative humidity, the Qinghai-Tibet Plateau has a typical and representative alpine grassland ecological environment.

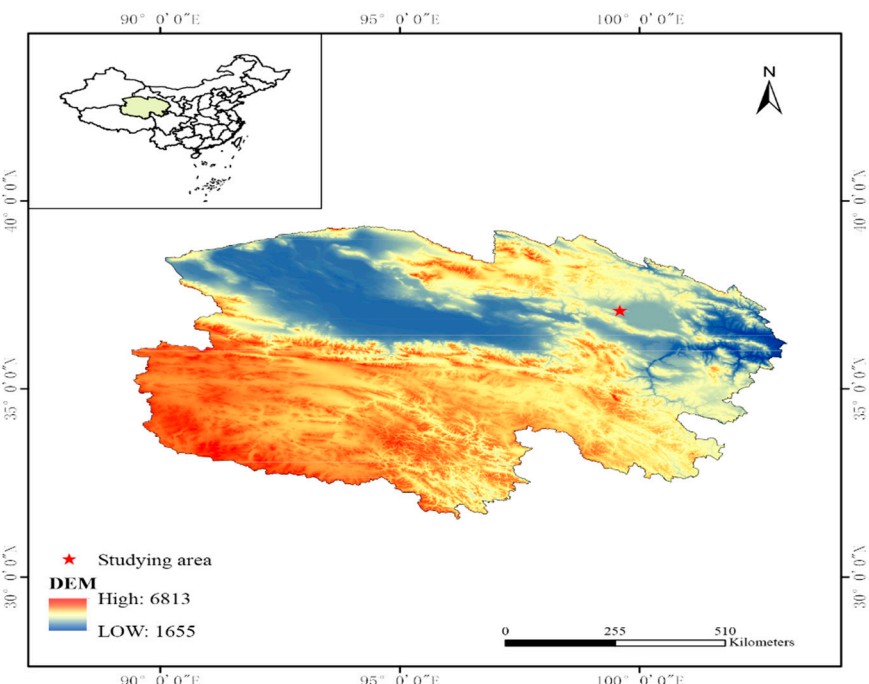

**Figure 1.** Distribution of study areas.

*2.3. Experimental Design*

According to the randomized group experimental design method, do plots $4 \times 3 = 12$ (3 m $\times$ 5 m), as shown in Figure 2, sample spacing 1 m [25]. The original data of the plot showed that the soil texture was calcium chestnut soil, with total k = 2.14%, total $p$ = 0.10%, total $n$ = 0.27%, and pH = 7.64. The seeds were sown at a rate of 133 kg/hm$^2$ in June 2020. The biofertilizer was mixed thoroughly with alfalfa seeds at the ratios of 0 kg/hm$^2$ (A1), 25 kg/hm$^2$ (A2), 50 kg/hm$^2$ (A3), and 75 kg/hm$^2$ (A4) before sowing (Table 1), so that the surface of each seed was evenly coated in the solution, then dried in a cool place and sown within 12 h. The coated seeds were sown in trenches 5 cm deep at 20 cm intervals, and the mixture was spread evenly into the trenches, covered, and stepped on. Samples were taken on 11 September 2020, using the random sampling method. Using the digging method, the plants were removed from the ground using a shovel. Then, adopting the cutting method, the ground part of the plant was cut on the ground, and plant roots and soil samples from 0–20 cm were collected. Bulk soil was shaken off the root system, and the adhering soil (0–5 mm) on the roots was collected as rhizosphere soil [26,27]. The intact rhizosphere root system and soil samples were immediately packed in sterile sealed bags and brought back to the laboratory at a low temperature (car refrigerator). The intact root system was used for laboratory observations, and the soil samples were divided into two parts, one stored at −80 °C for molecular analysis and the other air-dried and sieved and brought back to the laboratory in self-sealing bags for determination of soil physicochemical properties.

**Table 1.** Description of seed dressing dose.

| Plants | Mixed Bacteria Gradient | Repetition | Numbering |
|---|---|---|---|
| Beilin 201 | 0 | 3 | A1 |
| | 25% | 3 | A2 |
| | 50% | 3 | A3 |
| | 75% | 3 | A4 |

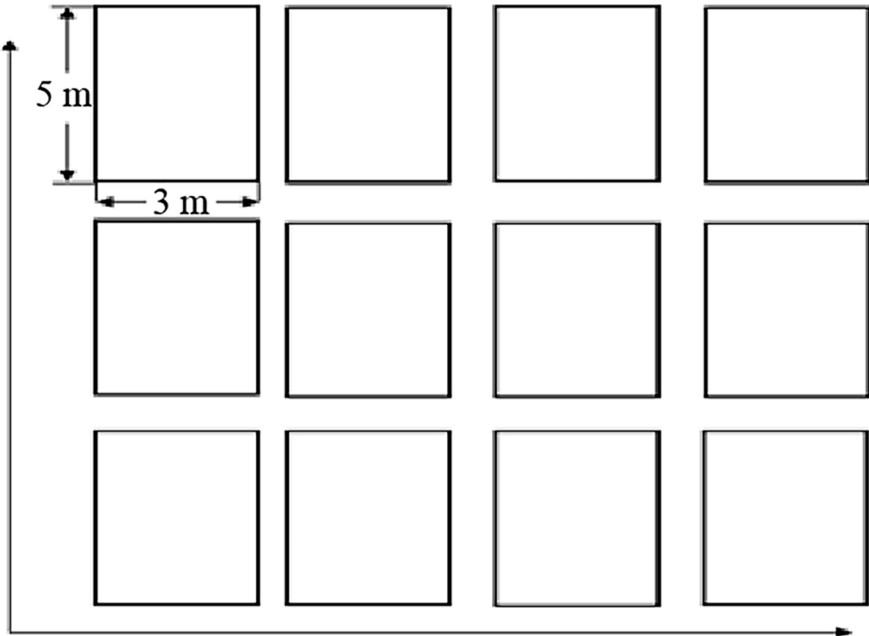

**Figure 2.** Schematic diagram of sample plot distribution sampling.

*2.4. Measurement Index and Method*

2.4.1. Root Traits and Soil Physicochemical Index

The intact root systems brought back to the laboratory were first separated from the attached root nodules, and laboratory tools were applied to measure root length, root diameter, and maximum root width to compare the effects of biofertilizer on root characteristics.

Soil environmental parameters (soil temperature, soil moisture, soil conductivity) were measured with a FieldScout TDR 350 soil moisture meter (Spectrum Technologies, Aurora, IL, USA) [11,28]. Triplicate measurements were conducted on each sample using a 7.60 cm probe. Physicochemical properties were determined according to previously published protocols [29,30]. Briefly, total nitrogen content was determined by the Kjeldahl method, nitrate nitrogen content was determined by the potassium chloride-hydrazine sulfate reduction method, and soil organic carbon content was determined by the potassium dichromate-sulfate volumetric method.

2.4.2. DNA Extraction and Amplicon Sequencing of 16S rRNA Gene and ITS rRNA Gene

Total DNA was extracted from 0.25 g rhizosphere soil using the Mobio Power Soil DNA isolation kit (Mobio Laboratories, Carlsbad, CA, USA) according to the manufacturer's instructions. To obtain the rhizosphere soil, the gramineous roots were washed 3 times with PBS buffer after excess soil was shaken from the roots. The suspensions were pooled and centrifuged, and the resulting sediment pellets were identified. Extracted DNA was amplified using the 16S rRNA universal prime set, 515 forward (5′-GTGCCAGCMGCCGCGGTAA-3′) and 806 reverse (5′-GGACTACHVGGGTWTCTAAT-3′), that targeted the V4 hypervariable regions of the prokaryotic 16S rRNA genes and were supplemented with sample-specific barcodes at both 5′ ends. Polymerase chain reaction (PCR) amplification was carried out in a 50 μL reaction system containing 1 μL of template DNA (20–30 ng/μL), 1 μL of both 10 uM forward and reverse primers, 1 μL Taq DNA Enzyme (TaKaRa), 25 μL 10 × PCR buffer, 1 μL dNTP mixture, and 20 μL ddH$_2$O. The thermal cycle conditions were as follows: initial denaturation at 95 °C for 3 min, followed by 30 cycles for 15 s at 95 °C, 15 s at 52 °C, 45 s at 72 °C, final extension at 72 °C for 5 min, and then held at 4 °C. Fungal genomic DNA was extracted using the Fast DNA spin kit for soil (MP Biomedical LLC, Irvine, CA, USA) according to the manufacturer's instructions. gITS7 (50-GTGARTCATCGARTCTTTG-30) and ITS4 (50-TCCTCCGCTTATTGATATGC-30) were used as primers to amplify the ITS region. The amplicons were separated on a 1% agarose

gel and then further recycled using Gel Extraction Kit (D2500-02, Omega BioTek, Norcross, GA, USA). The purified products were quantified using Nanodrop 2000 spectrophotometer (Nanodrop Technologies, Wilmington, DE, USA) and then pooled in an equimolar ratio for library construction. High-throughput sequencing was performed based on the Illumina Miseq sequencing platform (Magigene Biotech, Guangzhou, China).

*2.5. Processing of the Sequencing Data*

The 16S rDNA and ITS rRNA gene sequencing data were analyzed via Galaxy pipeline (http://mem.rcees.ac.cn:8080 (accessed on 21 March 2021), which consisted of an integrated series of bioinformatics tools [31]. Quality control procedures were in line with our earlier study (Wang et al., 2021). Low-quality sequences with an average quality score of <20 and short sequences with a length less than 200 bp were removed. Sequences were clustered into operable taxonomic units (OTUs) using UPARSE with a sequence similarity threshold of 97% to obtain OUT tables for subsequent microbial diversity analysis [32]. Soil chemical property data were subjected to a one-way analysis of variance (One-way ANOVA) using SPSS 25.0, and the least significant difference (LSD) method was used to compare the significance of differences between treatments. SigmaPlot 12.5 and Origin 2019 were used to plot graphs. Effects were considered significant if $p < 0.05$.

**3. Results**

*3.1. Analysis of Root System Characteristics after Different Gradient Treatments*

The influence of the root system on the plant is crucial; the root length and root diameter of the root system represent its groundwardness, and the number of lateral roots and root width of the root system symbolize waterwardness and chemotaxis. We found that the biofertilizer treatment increased the root length and root diameter of alfalfa, but the effect was not significant ($p < 0.05$). Figure 3 shows camera photos of the alfalfa root system treated with biofertilizer. Similarly, we obtained photographs of the root nodules shown in Figure 4 by separating the root system from the root nodules and found that alfalfa root nodules exhibited different shapes and colors. The main shapes exhibit coral, spherical, block, and rod shapes, while the colors are mainly white (ineffective rhizomes) and pink (effective rhizomes).

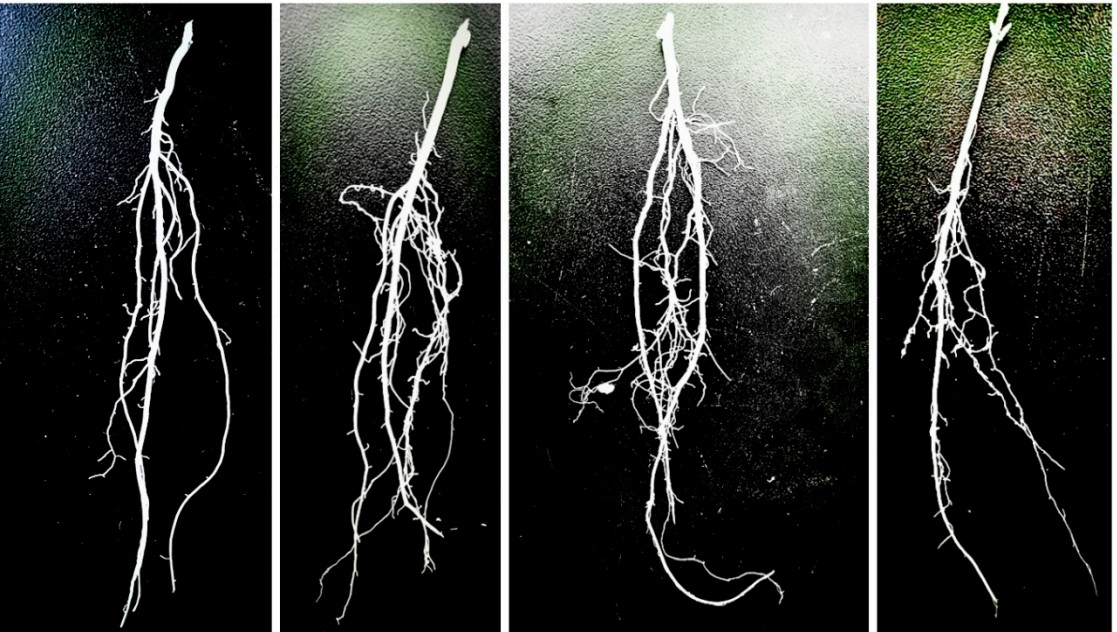

**Figure 3.** Photographs of the root system of the biofertilizer gradient. Note: The photos are A1, A2, A3, and A4 from left to right.

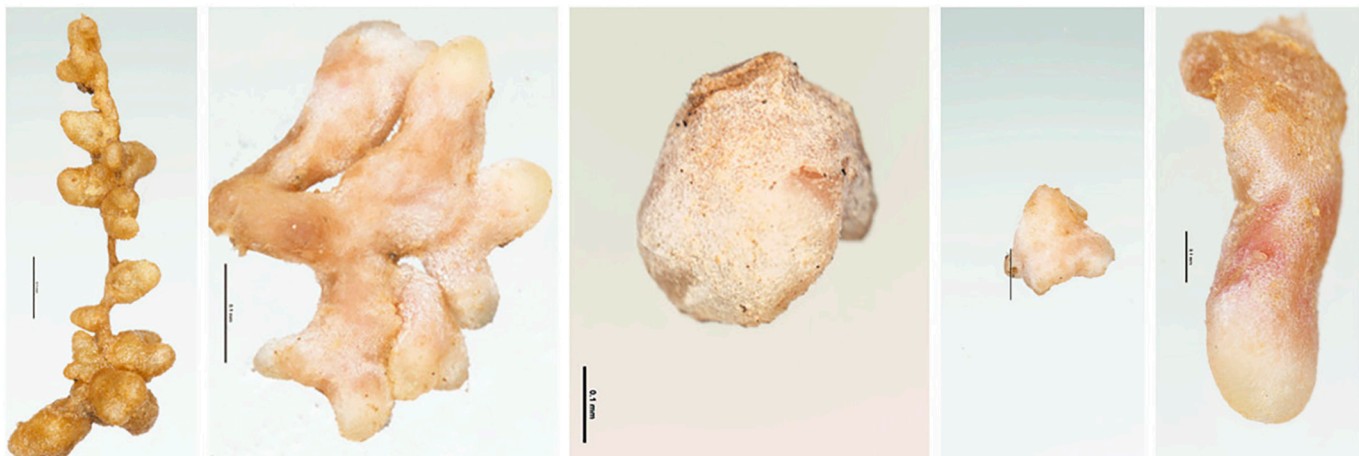

**Figure 4.** Photograph of alfalfa root tumor shape. Note: From left to right, ineffective rhizoma, effective rhizoma coral-like, effective rhizoma globular, effective rhizoma lumpy, and effective rhizoma rod-like.

From Table 2, it can be seen that the appropriate amount of biofertilizer can increase the number of lateral roots and root width of the alfalfa root system. The maximum root width of alfalfa in the A3 treatment reached 12.433 cm, and the difference between root width with no biofertilizer treatment (A1) and A4 treatment was significant ($p < 0.05$).

**Table 2.** Root system characterization.

| Group | Main Root Length/cm | Main Root Diameter/mm | Lateral Root Number | Max. Root Width/cm | Nodule Number |
|---|---|---|---|---|---|
| A1 | 31.89 ± 1.108 a | 5.06 ± 0.248 a | 3.27 ± 0.656 a | 10.98 ± 1.341 ab | 3.33 ± 0.333 b |
| A2 | 33.43 ± 0.653 a | 5.41 ± 0.240 a | 3.00 ± 0.577 a | 10.20 ± 0.327 ab | 4.33 ± 0.881 ab |
| A3 | 34.19 ± 1.852 a | 5.49 ± 0.068 a | 3.60 ± 0.305 a | 12.43 ± 0.993 a | 6.00 ± 0.577 a |
| A4 | 30.93 ± 1.326 a | 5.01 ± 0.658 a | 2.60 ± 0.346 a | 8.55 ± 0.427 b | 4.00 ± 0 b |

Note: different lowercase letters a and b represent significant differences between different treatments in the same column.

Root nodules are the key to nitrogen fixation in alfalfa, and the amount of root nodules affects how much free nitrogen the plant can fix and synthesize nitrogenous compounds for use by legumes. The analysis of the measurements revealed that the application of biofertilizer significantly increased the number of root nodules in the alfalfa root system (Table 2).

### 3.2. Analysis of Soil Physical and Chemical Properties

It can be seen in Table 3 that the soil moisture at 7.6 cm is 15.433–16.189%, with relatively high humidity. The contents of total nitrogen, nitrate nitrogen, and organic carbon were between 1161–1575 mg/kg, 30.5–34.74 mg/kg, and 1.463–2.183%, respectively. Compared with treatment A1, treatment A1 increased soil moisture but decreased electrical conductivity, total nitrogen, nitrate nitrogen, and organic carbon. The organic carbon and total nitrogen contents were significantly different due to the gradient of microbial fertilizer ($p < 0.05$).

**Table 3.** Basic physical and chemical properties of soil.

| Sample | Soil Moisture | Soil Conductivity (μS/cm) | TN (%) | Nitrate Nitrogen (mg/kg) | TOC (%) |
|---|---|---|---|---|---|
| A1 | 15.43 ± 2.46 a | 0.12 ± 0.04 a | 0.16 ± 119.41 b | 34.74 ± 4.35 a | 2.08 ± 0.07 b |
| A2 | 15.13 ± 2.04 a | 0.08 ± 0.03 a | 0.14 ± 6.51 ab | 32.61 ± 2.36 a | 1.65 ± 0.38 ab |
| A3 | 16.19 ± 0.85 a | 0.10 ± 0.02 a | 0.15 ± 54.24 b | 34.11 ± 0.70 a | 2.18 ± 0.26 b |
| A4 | 14.99 ± 0.91 a | 0.09 ± 0.01 a | 0.12 ± 196.27 a | 30.50 ± 3.10 a | 1.46 ± 0.35 a |

Note: different lowercase letters a and b represent significant differences between different treatments in the same column.

### 3.3. Soil Microbial Community Composition and Structure Analysis

By analyzing the number of unique and shared fungal OTUs (Figure 5a), we found that 12 fungal OTUs were shared by all 4 treatments. Additionally, 46 OTUs were common to A1 and A3, indicating that the fungal community structures of the 2 groups were highly similar. The number of unique OTUs for A1, A2, A3, and A4 were 186, 220, 222, and 202, respectively, accounting for 21.18%, 25.06%, 25.28%, and 23.01% of the total OTUs among all treatments, which was equivalent to that of no biofertilizer treatment, and the biofertilizer treatment increased the number of OTU fungi by 4.1%.

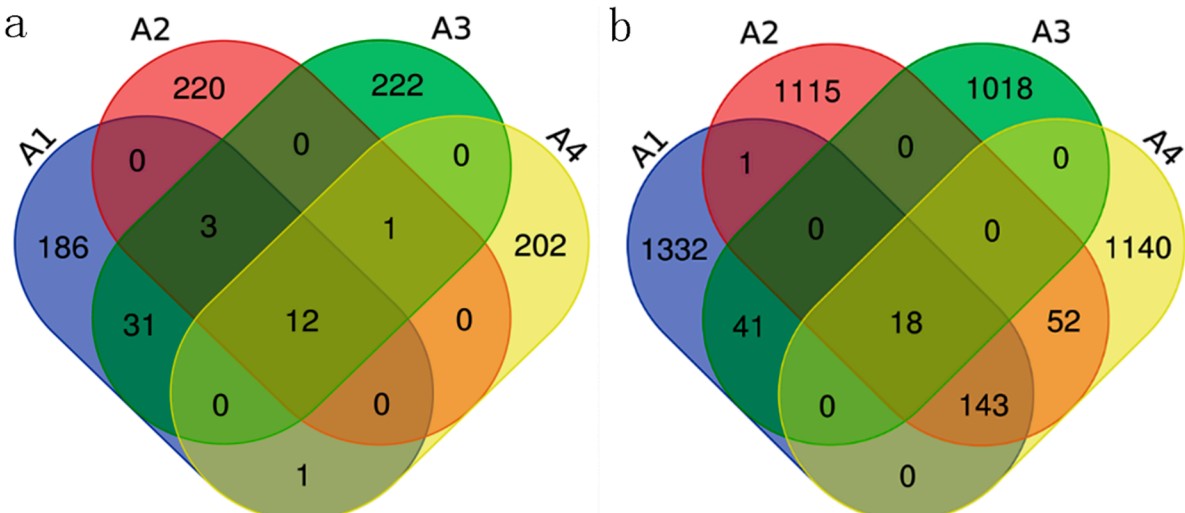

**Figure 5.** Sample Venn diagram. Note: (**a**) is the fungal Venn Diagram, (**b**) is the bacterial Venn Diagram.

Of the 4860 bacterial OTUs in total among the 4 treatments, only 18 OTUs were shared by all treatments (Figure 5b). The number of unique OTUs for A1, A2, A3, and A4 were 1332, 1115, 1018, and 1140, respectively, accounting for 27.41%, 22.94%, 20.95%, and 23.46% of the total OTUs, which was equivalent to the treatment without biofertilizer (A1). We also found that the biofertilizer treatment reduced the number of bacterial OTUs by 6.46%, indicating that the application of biofertilizer could reduce the species and diversity of soil microorganisms.

Through sequencing, it was found that the bacterial community structure of alfalfa rhizosphere soil displayed a high diversity at the genus and phylum levels under the biofertilizer gradient (Figure 6). After species classification of the OTUs, the community structure of the four biofertilizer gradients was firstly analyzed at the phylum level. As shown in Figure 6a, at the phylum level, fungal sequences mainly belonged to Ascomycota, Basidiomycota, and Unclassified. Among them, one phylum was not identified. Ascomycota and Basidiomycota were the two most abundant soil fungal phyla in the four treatments. Ascomycota was the most abundant fungal phylum in the 50% biofertilizer treatment (A3), accounting for 95.5%. The highest abundance of Basidiomycota was observed in the 25% treatment (A2), accounting for 6.35%. The results showed that the composition of the microbial community was different under the different biofertilizer mixtures.

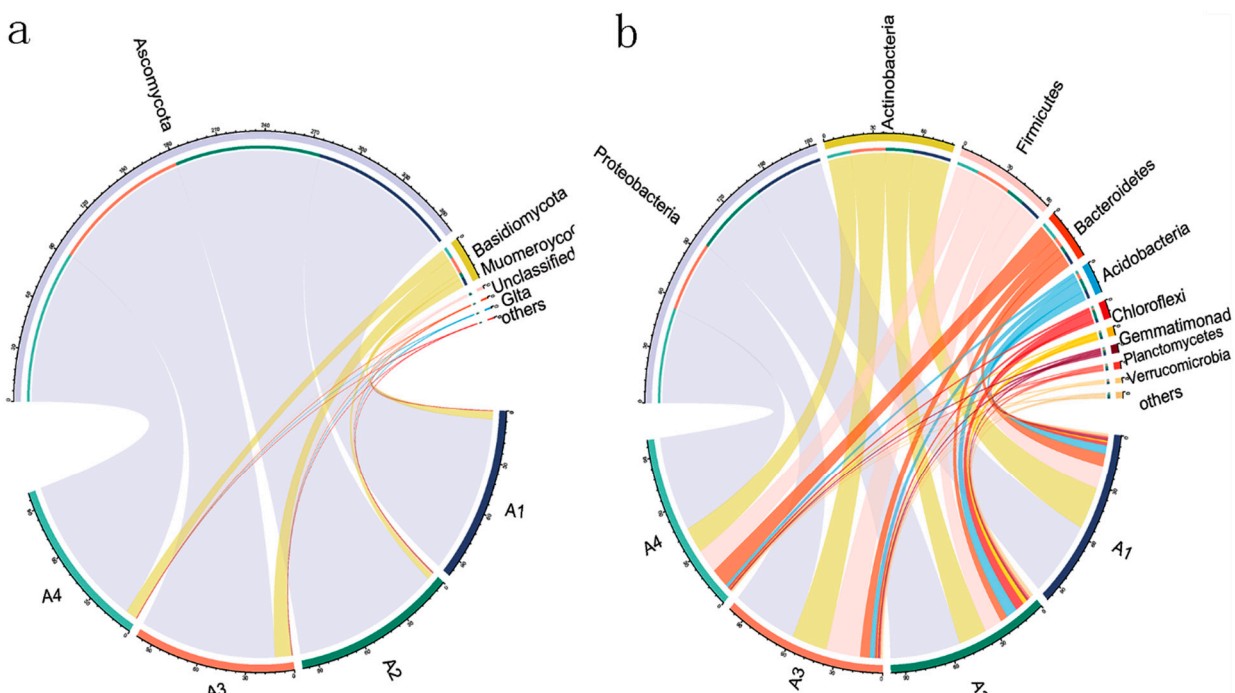

**Figure 6.** Soil microbiota horizontal community composition and chord diagram. Note: (**a**) represents fungal "phylum level", (**b**) represents bacterial "phylum level".

There was also great diversity at the phylum level among bacteria (Figure 6b) and included Proteobacteria, Actinobacteria, Firmicutes, Bacteroides, Acidobacteria, Uncategorized, Gemmatimonadetes, Planctomycetes, Chloroflexi, Verrucomicrobia, etc. Through comparative analysis, it was found that Proteobacteria, Actinobacteria, and Firmicutes were the most abundant bacterial phyla within the four biofertilizer treatments. Proteobacteria accounted for 54.28% when the biofertilizer gradient was 75% (A4), while the Actinobacteria phylum (actinobacteria) accounted for 24.62% in the no treatment (A1) condition. Acidobacteria, Uncategorized, Gemmatimonadetes, Planctomycetes, and Chloroflexi were all influenced by the biofertilizer, displaying a significant negative correlation with an increase in the percentage of the biofertilizer gradient.

The abundance heat maps (Figure 7) are based on the similarity clustering of the classification information of the top 10 species with the highest abundance and compare the abundance differences of genera among soil microorganisms along the biofertilizer gradient. As can be seen from Figure 7a, among the bacteria, Arthrobacter, Unclassified, Enterobacter, and Gp6 had the highest abundances in the 0% treatment. Pseudomonas, Acinetobacter, and Gp6 had the highest abundances in the 25% treatment, while Arthrobacter, Exiguobacterium, and Enterobacter had the highest abundances in the 50% treatment. Flavobacterium, Rhizobium and Sphingomonas were the highest in the 75% treatment. It can be seen that the abundance of some genera varied between treatments.

As shown in Figure 7b, the abundance of some fungal genera also varied among the different treatment levels. For example, Didymella had its highest abundance in the 0%, 50%, and 75% biofertilizer treatments and the lowest abundance under the 25% treatment. The abundance of Gibberella was lowest in the 0% treatment but increased with increasing biofertilizer gradient.

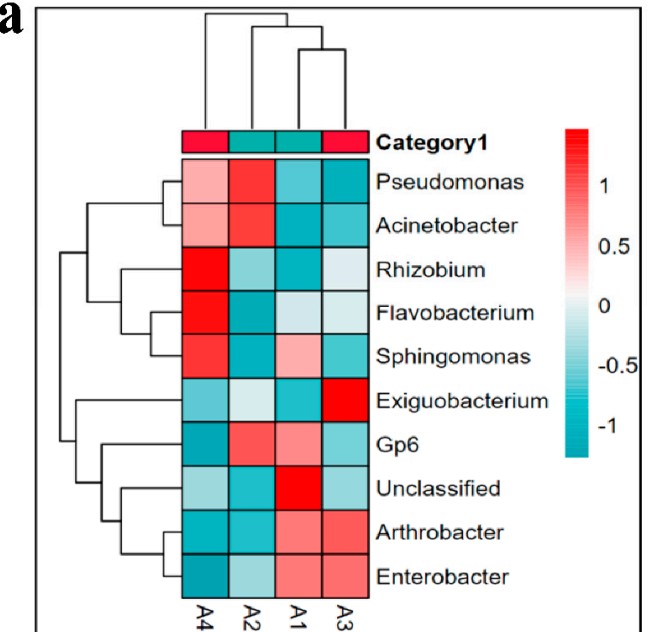
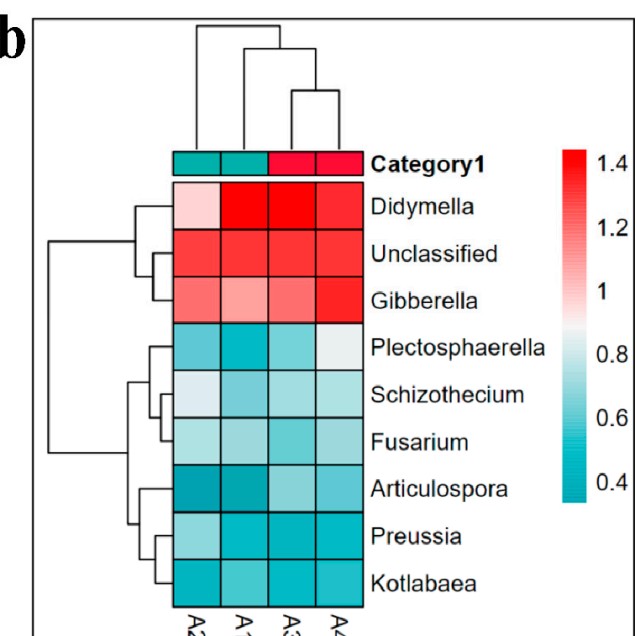

**Figure 7.** Abundance of community composition at the soil microbiota genus level. Note: (**a**) represents bacterial abundance heat map, (**b**) represents fungal abundance heat map

*3.4. Biome Diversity Analysis*

The $\alpha$ diversity of the rhizosphere soil microbial community was compared with four different microbial doses of alfalfa. Shannon index, Inv_Simpson index, Observed_richness index, and Pielou_evenness index were applied to show the $\alpha$ diversity of the rhizosphere soil microbial community. As shown in Table 4, the Shannon index and richness index (0Bserved_richness index) were the highest in the A1 (CK) treatment and the lowest in the A3 treatment. However, the Inv Simpson index and Pielou evenness index were positively correlated with the increase of bacteria gradient ($p < 0.05$). In fungal microbial diversity, both Shannon index and Inv_Simpson index showed an increasing trend with the dose of bacteria mixing, indicating that the bacteria mixing treatment increased the microbial and fungal diversity of alfalfa rhizosphere soil.

**Table 4.** Sample $\alpha$ diversity index.

| Microbe | Samples | Shannon | Inv_Simpson | Observed_Richness | Simpson_Evenness |
|---------|---------|---------|-------------|-------------------|------------------|
| Bacter | A1 | 4.798 ± 0.503 a | 22.359 ± 4.912 a | 2061.67 ± 451.23 a | 0.011 ± 0.001 a |
|  | A2 | 4.681 ± 0.661 a | 27.653 ± 14.308 a | 1901.00 ± 347.15 a | 0.014 ± 0.005 a |
|  | A3 | 4.346 ± 0.789 a | 28.078 ± 15.466 a | 1739.33 ± 435.12 a | 0.014 ± 0.006 a |
|  | A4 | 4.665 ± 0.209 a | 28.871 ± 5.978 a | 1761.67 ± 128.23 a | 0.016 ± 0.002 a |
| Fungi | A1 | 3.874 ± 0.551 a | 21.723 ± 11.377 a | 825.33 ± 79.55 a | 0.024 ± 0.011 a |
|  | A2 | 3.920 ± 0.633 a | 22.514 ± 8.982 a | 771.67 ± 159.91 a | 0.025 ± 0.008 a |
|  | A3 | 4.389 ± 0.127 a | 26.132 ± 5.357 a | 909.00 ± 60.85 a | 0.028 ± 0.005 a |
|  | A4 | 4.168 ± 0.157 a | 22.112 ± 1.632 a | 791.33 ± 82.49 a | 0.028 ± 0.004 a |

Note: different lowercase letters represent significant differences between different treatments in the same column.

*3.5. The Relationship between Soil Microorganisms and Environmental Factors*

In order to explore the correlation between the physical and chemical properties of the sample planting soil and species diversity, a Pearson correlation analysis was conducted, and the results are shown in Table 5. If the tightness of the relationship is determined to be above 0.7, the relationship is close. A relationship between 0.5 and 0.7 is general. The simple correlation coefficients in Table 5 show that there was no significant correlation among the components without fertilizer treatment, and after fertilizer treatment, there was a

very strong and significant positive correlation between species diversity indices ($p < 0.01$), and soil moisture and soil conductivity were significantly and positively correlated with the observed richness of species ($p < 0.05$). Pearson correlation analysis indicated that the diversity of the microbial community in the inter-rhizosphere soil of north forest 201 alfalfa was influenced by the physicochemical properties of the soil after the treatment with bacterial fertilizer.

**Table 5.** Correlation analysis of soil physical and chemical properties and species diversity.

| Correlation | Shannon | Inv_Simpson | Observed_Richness | Pielou_Evenness | Soil Moisture | Soil Conductivity | TN | Nitrate Nitrogen | TOC |
|---|---|---|---|---|---|---|---|---|---|
| Shannon | 1 | 0.866 | 0.973 | 0.997 * | 0.908 | 0.807 | 0.084 | 0.295 | 0.587 |
| Inv_Simpson | 0.807 ** | 1 | 0.728 | 0.899 | 0.576 | 0.403 | 0.571 | 0.733 | 0.103 |
| Observed_richness | 0.941 ** | 0.621 * | 1 | 0.954 | 0.980 | 0.921 | −0.147 | 0.067 | 0.757 |
| Pielou_evenness | 0.985 ** | 0.827 ** | 0.902 ** | 1 | 0.875 | 0.762 | 0.155 | 0.363 | 0.527 |
| Soil Moisture | 0.502 | 0.427 | 0.595 * | 0.427 | 1 | 0.980 | −0.342 | −0.134 | 0.873 |
| Soil conductivity | 0.413 | 0.132 | 0.596 * | 0.434 | 0.45 | 1 | −0.521 | −0.327 | 0.952 |
| TN | −0.176 | −0.293 | −0.039 | −0.195 | 0.1 | 0.154 | 1 | 0.977 | −0.758 |
| nitrate nitrogen | −0.059 | −0.168 | −0.006 | −0.064 | 0.021 | −0.071 | 0.741 ** | 1 | −0.601 |
| TOC | −0.291 | −0.479 | −0.077 | −0.335 | 0.119 | 0.092 | 0.788 ** | 0.592 * | 1 |

Note: The lower left corner is the correlation data after bacterial fertilizer mixing; The correlation data of CK (without bacterial fertilizer treatment) is in the upper right corner. **. At 0.01 level ($p < 0.01$), the correlation was significant. *. At 0.05 level ($p < 0.05$), the correlation was significant.

In order to study the correlation between the physical and chemical properties of soil planted with samples and the distribution of soil microbial community, the first three dominant phyla of bacteria—Proteobacteria, Actinobacteria, Firmicutes and the first two dominant phyla of fungi—Ascomycota and Basidiomycota, were selected. Pearson correlation analysis was also conducted, and the results are shown in Table 6. In the unmixed treatment, Ascomycota and Basidiomycota showed a significant negative correlation ($p < 0.01$). There was a significant positive correlation between organic carbon and Firmicutes ($p < 0.05$). After treatment with fungus fertilizer, Proteobacteria was negatively correlated with nitrate nitrogen ($p < 0.05$). Actinobacteria were positively correlated with nitrate nitrogen, organic carbon, and total nitrogen ($p < 0.01$) but negatively correlated with Proteobacteria ($p < 0.05$). Firmicutes were positively correlated with soil water content ($p < 0.05$). The results showed that the treatment of bacterial fertilizer enhanced the correlation between soil physical and chemical properties and soil microbial community distribution.

**Table 6.** Correlation analysis of soil physical and chemical properties and other factors.

| Correlation | TN | Nitrate Nitrogen | TOC | Soil Moisture | Soil Conductivity | Ascomycota | Basidiomycota | Proteobacteria | Actinobacteria | Firmicutes |
|---|---|---|---|---|---|---|---|---|---|---|
| TN | 1 | 0.977 | −0.758 | −0.342 | −0.521 | 0.008 | −0.001 | −0.120 | 0.735 | −0.732 |
| nitrate nitrogen | 0.741 ** | 1 | −0.601 | −0.134 | −0.327 | 0.221 | −0.214 | −0.329 | 0.863 | −0.569 |
| TOC | 0.788 ** | 0.592 * | 1 | 0.873 | 0.952 | 0.647 | −0.652 | −0.557 | −0.115 | 0.999 * |
| Soil Moisture | 0.054 | −0.197 | 0.100 | 1 | 0.980 | 0.937 | −0.939 | −0.892 | 0.385 | 0.891 |
| Soil conductivity | 0.265 | 0.038 | 0.238 | 0.356 | 1 | 0.850 | −0.853 | −0.785 | 0.195 | 0.963 |
| Ascomycota | −0.266 | −0.128 | −0.427 | 0.121 | 0.006 | 1 | −1.000 ** | −0.994 | 0.683 | 0.676 |
| Basidiomycota | 0.280 | 0.075 | 0.407 | −0.040 | −0.018 | −0.982 ** | 1 | 0.993 | −0.678 | −0.681 |
| Proteobacteria | −0.381 | −0.609 * | −0.506 | −0.257 | −0.431 | 0.276 | −0.179 | 1 | −0.761 | −0.589 |
| Actinobacteria | 0.718 ** | 0.855 ** | 0.737 ** | 0.217 | 0.224 | −0.080 | 0.055 | −0.703 * | 1 | −0.076 |
| Firmicutes | 0.118 | −0.233 | 0.050 | 0.620 * | 0.299 | −0.362 | 0.477 | −0.110 | −0.072 | 1 |

Note: The lower left corner is the correlation data after bacterial fertilizer mixing; The correction data of CK (without bacterial fertilizer treatment) is in the upper right corner. **. At 0.01 level ($p < 0.01$), the correlation was significant. *. At 0.05 level ($p < 0.05$), the correlation was significant.

## 4. Discussion

Root growth and development, to some extent, reflect the growth of the plant [33]. The results of this experiment confirm that the application of biofertilizer in the Tibetan plateau can increase the root length and rhizome diameter of forage grasses and increase the uptake of water and minerals from the soil by alfalfa forage grasses, and can better

supply forage plants with the required nutrients. Recent studies [34] have pointed out that legumes are sensitive to nitrogen fertilizer and are quickly absorbed and utilized by plants when easily available nitrogen fertilizer is present. Meanwhile, there are more and more studies [1,35] on rhizosphere nitrogen fixation symbiosis in legume forages, and this study found that biofertilizers can increase the number of rhizomes in alfalfa forage roots, which fits with common sense.

Changes in soil's physical and chemical properties directly affect soil quality, and soil quality is one of the important factors restricting agricultural production [36,37]. Therefore, in recent years, the number of studies on soil's physical and chemical properties has gradually increased. Some researchers have carried out studies on the relationship between soil's physical and chemical properties in different land types (such as forest land, farmland, grassland, etc.) [38–40], while others have studied the effects of corn stalk composting on soil's physical and chemical properties [41]. It was found that the application of bio-organic fertilizers increased soil organic matter content as well as soil enzyme activity [42]. Bio-organic fertilizers contain a large amount of organic matter and beneficial soil microorganisms [43,44], which can produce a vast amount of organic acids through their metabolic activities, as well as continuously release late-effective nitrogen, phosphorus, and potassium from the soil, which can effectively improve the physical and chemical properties of the soil and increase soil fertility [45,46]. In the current study, the biofertilizer treatment increased soil moisture and organic carbon content as compared to the unfertilized treatment (A1), and the organic carbon and total nitrogen content were significantly different ($p < 0.05$) due to the biofertilizer gradient. This indicated that the biofertilizer mix improved the physicochemical properties of the soil, which is consistent with the findings of previous studies [37,47,48].

Soil microorganisms are an important part of the soil ecosystem and are the driving force of material transformation in the soil. The structural composition of their communities and their diversity reflect the quality of the soil to a certain extent, and they are also the key to overcoming succession barriers and other soil barrier factors [49]. It has previously been found that plant rhizosphere soil microbial communities are dominated by bacteria [50], which is consistent with the conclusion of the current study, based solely on the number of phyla/genera, with the number of bacteria in the rhizosphere soil of alfalfa with different biofertilizer treatments being significantly higher than that of fungi. The soil bacterial (16S rRNA gene) OTU clustering analysis of this study revealed that the 4 biofertilizer treatments of alfalfa shared only 18 OTUs, and the number of unique OTUs was negatively correlated as the mixing gradient increased, indicating that the application of biofertilizer affected microbial population changes. Studies have shown that microbial agents and bio-organic fertilizers can significantly improve the microenvironment around the roots of plants, regulate the types and numbers of bacteria, and promote the growth of beneficial bacteria [44,48]. We found that the biofertilizer treatment increased the relative abundance of Proteobacteria, Actinobacteria, Ascomycota, and Basidiomycota at the phylum level.

According to Hua et al. [51], the Simpson index is more sensitive to species evenness than the Shannon–Wiener index, while the latter is more sensitive to species richness. From the alpha indices analysis of this experiment, the Shannon (Shannon) and the richness (Observed_richness) indices of the rhizosphere microbial diversity of the four mixing treatments were found to be consistent, while the Simpson (Inv Simpson) and the evenness (Pielou evenness) indices were more closely related, which was consistent with the findings of Hua et al. (2022), all of which indicated that the mixing treatments affected the diversity of soil microorganisms.

Pearson correlation analysis was performed to investigate the correlation between the soil physicochemical properties and species diversity and the distribution of soil microbial communities [52] and showed that the treatment with biofertilizer enhanced the significant positive correlation between species diversity indices in this study. Soil moisture and soil conductivity were significantly and positively correlated with the observed richness of species ($p < 0.05$), while nitrate-nitrogen, organic carbon, and total nitrogen also

showed very strong and significant positive correlations. Proteobacteria and Actinobacteria both showed significant positive correlations with soil physicochemical properties ($p < 0.05$), while Actinobacteria showed a very strong negative correlation with Basidiomycota ($p < 0.05$), suggesting that there is a correlation between soil physicochemical factors and soil microbial diversity and community structure. This is in agreement with some previous studies that showed the community structure of plant rhizosphere microorganisms as being influenced by soil physicochemical properties [36,53–55].

## 5. Conclusions

To summarize, soil microorganisms are the driving force of soil ecosystems and constantly change the soil microenvironment. Vegetation affects the structure and diversity of soil microorganisms by affecting the soil environment. In this study, we explored the structure and diversity of the soil microbial community in the rhizosphere of alfalfa with four different mixing gradients of a biofertilizer in alpine grassland and found that there were similarities and differences in the structure and composition of the microbial community among the different treatments, indicating that the mixing treatments affected the structure and diversity of soil microbial community. At the same time, through Pearson correlation analysis, it was found that biofertilizer treatment increased the soil microbial community structure and diversity affected by major soil physical and chemical factors and improved the relative abundance of beneficial microorganisms. This experiment laid a foundation for the subsequent study of alfalfa rhizosphere soil microbial community distribution, continuous cropping effect, and related rhizosphere microorganisms in alpine grassland. Using biofertilizers can help to realize the potential of the planting model of grassland agricultural production and environmental friendship; it can also improve the foundation for improving the soil with excellent production and ecological properties. It is one of the measures to explore in artificial grassland planting and grassland ecological restoration. This experimental biofertilizer can improve efficiency and increase green agricultural development through biological nitrogen-fixing nitrogen and complementary nutrients.

**Author Contributions:** Y.Z., G.L. and Y.W. conceptualized and designed the experiments; K.M., G.L., X.Z., H.Y. and Y.Z. performed the experiments; Y.Z., H.Z. and X.J. analyzed the data; Y.Z. wrote the paper. All authors reviewed and agreed with the paper. All authors have read and agreed to the published version of the manuscript.

**Funding:** This research was supported by the general program of basic research of Qinghai Provincial Department of science and Technology (2021-zj-915), China in 2021.

**Data Availability Statement:** Data were deposited in the China National Microbiology Data Center (NMDC) with accession numbers SUB1651052286055. All other relevant data are available from the corresponding author on request.

**Conflicts of Interest:** The authors declare no conflict of interest.

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
