# Peer review of "Effects of Microbial Fertilizer on Soil Fertility and Alfalfa Rhizosphere Microbiota in Alpine Grassland"

_agronomy, doi:10.3390/agronomy12071722_

Round 1

Reviewer 1 Report

Some suggestions to improve the article.

The English and Grammar must be fully revised.

Abstract –

Add which hypothesis was tested. Add the experimental design used.

Correlation does not imply causation. I would rather change which results are added to the abstract. Which statistical model did you use to test the treatments?

Introduction

Line 40-42 Remove this sentence, it is not suitable or necessary – “As early as 1994, at the International Symposium 40 on Microbial Fertilizer, China's nodulation and nitrogen-fixing fertilizer technology of 41 Gramineae surpassed that of the United States, Canada, Australia and other countries”

Line 43-44 – Rephrase it. Biofertilizers can not decompose P and K elements. I understand what you mean, but the word might be released, or if you’re talking about microorganisms, it means that they can mineralize P and K in the soil.

Line 48-50 – Which region?

It is missing a hypothesis. Please add it clearer at the end of the introduction section, followed by the objectives of the research.

Material and methods

Line 68 – Please add more details about the N fixing fertilizer (e.g. types of strains)

Line 85- rate of 84 2000 g/mu. Please refer to the área based on the international standard, and journal requirements. I would suggest hectares.

Line 86 – Remove these A1, A2,… leave it as 0,25,50 … in the whole text do this change it is easier for anyone to read and understand it faster. Modified it in the tables and figures too.

Line 127 - The polymerse china reaction

You must add this to the article.

-There is missing information about the soil. Physical-chemical properties, especially soil texture (sand, clay and silt), CEC, P, and pH, among other indicators. You must add this to the article.

- Information regarding the fertilizer application. Including other fertilizers (chemical, organic…)

- Information about the harvesting of the alfalfa,  period of the experiment.

Results

Fig 3 and 4. You must make it clear which picture is associated with each treatment.

Table 2. Why A4 or 75g/mu reduced root length and nodules?

Table 5. It would be interesting to add pH to this analysis, as it has a strong influence on the microbial community.

Be careful when discussing the correlation of environmental factors with microbial communities, as correlation does not imply causation.

Discussion

301-309- Please add references to the statements you provided.

319-320 – Please revise the statement that biofertilizers contain beneficial soil microbes. 1 it can contain harmful ones too (e.g., pathogens); 2- does every biofertilizer are the same? So be specific with which one you’re talking about.  Focus on discussing related experiments that tested a similar biofertilizer you used in your trial.

Conclusions

Please add a hypothesis to your introduction, and then, solely based on your conclusion in your hypotheses tested. Please add, what advances your article brings to science and the knowledge of biofertilizer use.

Reviewer 2 Report

This manuscript provides valuable information about the effect of different doses of bacteria-mixing fertilizer on soil fertility and alfalfa rhizosphere microbiota and the relationship between microbial community structure and main physico-chemical properties of soil. The topic of this work closely falls within the aims and scope of the journal. The introduction is based on recent findings. The main objectives are clearly stated and supported by the literature review. In general section “Materials and Methods” is well written and documented. Tables (2-4) need some modifications in terms of the definition of significant differences among treatments and the position of the note (Tables 5-6). A few modifications are also recommended for Table 3 regarding the format of the data and the units of soil properties. The outcomes in section “Discussion” are well explained and the discussion of results focuses on the main points while justification of the findings is well supported by relative references.  Based on the above the manuscript can be considered for publication in this journal. Please see attached file for suggestions and recommendations. 

Author Response

Response to Reviewer 2 Comments

Q1: Tables (2-4) need some modifications in terms of the definition of significant differences among treatments and the position of the note (Tables 5-6).

R1: Thank you very much for your valuable comments. I have revised them in the revised manuscript according to your comments and marked them with red font.

Q2. A few modifications are also recommended for Table 3 regarding the format of t he data and the units of soil properties. 

R2: First of all, thank you very much for your valuable comments. Secondly, for the revision of Table 3, I have revised it according to the your suggestions and marked the red font in the revised manuscript.

Round 2

Reviewer 1 Report

Dear authors,

Thank you for the editions performed, however, there are still some issues to solve before acceptance. 

This part needs to be edited

Abstract 

Based on biological fertilizers, the trend of chemical fertilizer is gradually replaced, and 12 new biological fertilizers can improve soil fertilizer and soil microorganisms. This experiment is 13 based on the bean federy sacrifice (Medicago sativa cv. Beilin 201) as the research object. Purple flow- 14 ers' influence of soil micro -environment

What is bean federy sacrifice? I googled it and could not find anything related to it. Medicago sativa is alfafa. 

This phrase is also very strange - Purple flowers' influence of soil micro -environment?? If you're talking about Medicago sativa just say alfafa over the text. 

The hypothesis is still missed in the introduction

Introduction

This phrase needs to be revised.

, It is found that bio-fertilizer can also promote the elements of microbial decomposition and potassium minerals to absorb or secrete growth hormones to stimulate the growth of crops

In the last paragraph of the introduction, you must clearly add a hypothesis

I would suggest a very deep English review prior acceptance.

Kind regards.
